# Position: Deployed Reinforcement Learning should be Continual

**Parnian Behdin** [* 1]   **Kevin Roice** [* 1]   **Golnaz Mesbahi** [1]

## Abstract

Reinforcement Learning (RL) has received increasing attention and adoption in real-world use cases. Most of these systems follow a train-then-fix paradigm, where trained agents do not learn while interacting with the world until performance degrades and retraining becomes necessary. In this position paper, we argue that deploying an agent that is incapable of optimality, but receives an evaluative reward signal, is inherently a continual RL problem. We identify four sources of non-stationarity after deployment that necessitate never-ending learning, and highlight why the best deployed agents never stop adapting. We analyze successful examples of continual RL in the real world, and present the community with the advantages and measures to move away from the current train-then-fix paradigm.

## 1. Introduction

Reinforcement Learning (RL) is learning from interaction (Sutton & Barto, 2018). Yet after RL agents are deployed in the real world, they typically stop learning. Policies are trained offline (through simulations, self-play, expert demonstrations or a combination of these) and then frozen upon deployment while the world continues to change. The environment's complexity far exceeds what any finite training phase can capture, and re-training becomes necessary (Dulac-Arnold et al., 2019). We call this the *train-then-fix* paradigm.

This train-then-fix paradigm pervades the history of RL. TD-Gammon excelled at backgammon through self-play, and was frozen when competing (Tesauro, 1995). AlphaGo defeated world champion Lee Sedol (Silver et al., 2016), OpenAI Five defeated Dota 2 world champions (Berner

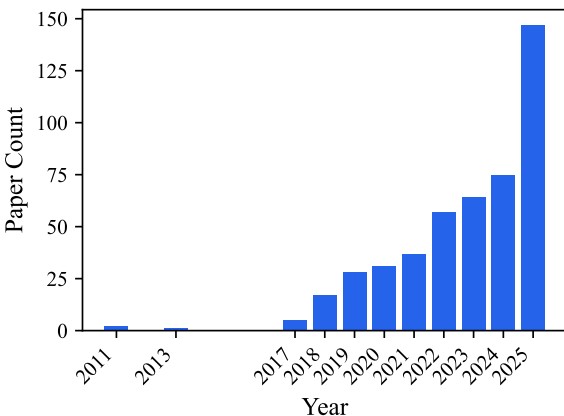

*Figure 1.* The number of papers on arxiv.org containing the words *continual reinforcement learning* in their title or abstract each year.

et al., 2019), and GT Sophy outperformed professional Gran Turismo drivers (Wurman et al., 2022). Deep RL has even controlled stratospheric balloons (Bellemare et al., 2020) and tokamaks (Degrave et al., 2022). In each case, policies were trained extensively offline and held fixed after deployment. While being landmark achievements for the field, these examples constrained the deployment problem to fit the train-then-fix paradigm. Environments were either stationary, easy to simulate accurately, or the deployment was brief or localized enough such that significant drifts from human knowledge and data during training did not emerge. This set the stage for extensive training pre-deployment in favor of learning continually from the observation stream after deployment.

Historically, we have been approximating continual learning problems as non-continual ones. Our methods would likely fail if deployed for extended periods, exposed to unforeseen distribution shifts, or asked to generalize beyond their training. The train-then-fix paradigm does not remedy the continual learning problem; it only delays it.

This is not a new observation (Hamadanian et al., 2022; Khetarpal et al., 2022). The Big World Hypothesis (Javed & Sutton, 2024) formalizes this insight that real-world environments exceed any agent's representational capacity. In deployment, resource constraints compound this challenge. Limited time, computation, and data may prevent agents from finding optimal policies even when they are representable in theory.

*Equal contribution, ordered alphabetically. [1] Alberta Machine Intelligence Institute (Amii), Edmonton, Canada. Correspondence to: Parnian Behdin <behdin@ualberta.ca>, Kevin Roice <kevin.roice@amii.ca>.

*Proceedings of the 43rd International Conference on Machine Learning*, Seoul, South Korea. PMLR 306, 2026. Copyright 2026 by the author(s).

The idea of blurring the line between training and deployment is not a new one either (Ring, 1994; Thrun, 1998). It reflects a view of learning as adaptation rather than solving a fixed problem (Barron et al., 2015; Abel et al., 2024).

Many deployed RL agents continue to receive evaluative feedback (via a reward signal) after deployment: recommendation systems observe user engagement, ride-sharing platforms track completed rides, and coding assistants measure suggestion acceptance rates. When such evaluative feedback is available, and the deployment environment exceeds the agent's representational capacity or the available time or compute resources, there is little justification to not leverage this signal for continued adaptation.

We call such settings *measurable* deployment: where the agent is constrained in its capacity and resources, but an evaluative reward signal remains available after deployment.

In this paper, we argue that **measurable deployment is a continual reinforcement learning *problem*, and therefore continual learning *solutions* should be utilized in deployed models**. With theoretical foundations maturing, academic interest surging (Figure 1), and successful industry deployments emerging, it is timely to advocate for this paradigm shift.

## 2. Background

### 2.1. Continual Reinforcement Learning

Abel et al. (2023) define the continual reinforcement learning (CRL) problem as: "*An RL problem is an instance of CRL if the best agents never stop learning*". This definition stands in contrast to the deployed RL examples in Section 1, where the agent searches for a policy, after which learning ceases and the fixed policy is deployed.

**The Problem with Traditional RL Foundations.** This norm of fixing the agent after a training phase comes in part from the mathematical formalization of the RL problem. The traditional Markov Decision Process (MDP; Bellman, 1957; Puterman, 2014) formalism of the agent-environment interaction fails to capture the never-ending nature of CRL. An MDP is represented by the tuple $\langle \mathcal{S}, \mathcal{A}, P, R \rangle$, where $\mathcal{S}$ and $\mathcal{A}$ are the state and action spaces. On each discrete timestep $t$, the agent selects an action $A_t \in \mathcal{A}$ in state $S_t \in \mathcal{S}$, the environment transitions to a new state $S_{t+1}$, emitting a scalar reward $R_{t+1}$. The MDP objective defines optimality with respect to transition and reward functions, $P : \mathcal{S} \times \mathcal{A} \mapsto \Delta(\mathcal{S})$ and $R : \mathcal{S} \times \mathcal{A} \mapsto \mathbb{R}$, for which an optimal policy, $\pi^\star : \mathcal{S} \mapsto \Delta(\mathcal{A})$, exists as a fixed point of the Bellman optimality equation (Puterman, 1990). This notion of converging to $\pi^\star$ implies a terminal point: once found, the policy is to be deployed indefinitely without further learning. Such a framing treats learning as a means to an end rather than an ongoing process (Abel et al., 2024). Additionally, the MDP assumptions of ergodicity, stationarity, and the ability to reset or revisit states rarely hold in deployment.

**The History Process Formalism.** To address these limitations, recent works propose the *history process* as an alternative mathematical foundation (Bowling et al., 2023; Abel et al., 2023). In this formalism, an environment $e$ is a function from finite-length histories and actions to a distribution over a finite observation space, $e : \mathcal{H} \times \mathcal{A} \mapsto \Delta(\mathcal{O})$, where the observation space is denoted by $\mathcal{O}$[1], action space denoted by $\mathcal{A}$, and the history set $\mathcal{H} = \bigcup_{n=0}^{\infty} (\mathcal{A} \times \mathcal{O})^n$ is the space of all possible finite sequences of action-observation pairs. The reward function is over such pairs, $R : \mathcal{O} \times \mathcal{A} \mapsto \mathbb{R}$. Following Elelimy et al. (2025), we define a policy $\pi : \mathcal{S} \mapsto \Delta(\mathcal{A})$ as a mapping from the agent's state representation to a distribution over actions. $S_t \in \mathcal{S}$ is now the agent's compression of its history, not to be confused with state from the MDP formalism. The set of all policies representable by the agent is denoted by $\Pi$. The agent's learning rule, $\sigma : \mathcal{H} \mapsto \Delta(\Pi)$, maps histories to a distribution over the policy set.

The history process makes minimal assumptions about the environment. Critically, it does not assume that agents can reset the environment or revisit previous histories. Once a history $h_t \in \mathcal{H}$ has occurred, the agent cannot ever pass $h_t$ back into the environment again. It can never exactly revisit the situations it was in before, and future interactions take the form $e(h_t \cdot h, a)$, where $\cdot$ denotes concatenation. Note that these formalisms include both episodic and continuing settings, as both can be continual learning problems.

### 2.2. Continual Learning Problems vs. Solutions

It is important to distinguish continual learning *problems* (problem settings where never-ending adaptation is useful) from continual learning *solutions* or *algorithms* (methods designed to address these problems) (Khetarpal et al., 2022). Continual learning algorithms typically address challenges with function approximators within the agent, such as catastrophic forgetting (McCloskey & Cohen, 1989), maintaining plasticity (Dohare et al., 2024), and balancing stability with adaptability (Mermillod et al., 2013). However, the existence of these algorithmic challenges does not define what makes a problem a continual learning problem. As Abel et al. (2023) emphasize, a problem is an instance of CRL based on the environmental characteristics that necessitate never-ending learning, not on the algorithmic difficulties of implementing such learning. We examine the characteristics of real-world deployment that necessitate never-ending learning in Section 3.

---

[1]These observations are not necessarily Markovian, allowing history processes to describe both MDPs and Partially Observable MDPs (Monahan, 1982; Cassandra et al., 1994).

## 2.3. Deployment

Throughout this paper, we use the term *deployment* to refer to integrating a trained policy into its intended operating environment where it actively makes decisions in the real-world. Deployment marks the transition from offline development to operational use, where the agent's performance is truly tested and its value realized (IBM, 2024).

# 3. Measurable Deployment is a Continual Reinforcement Learning Problem

Not *all* deployed systems require continual learning. Schaeffer et al. (2007)'s checkers engine has provably solved the game, and learning will not improve its win rate. Fixed policies suffice when the environment lies within the agent's representational capacity and available resources.

Many deployed RL systems operate in the big world regime, where the environment's complexity exceeds the agent's capacity and resources. In such settings, an optimal policy may be unrepresentable, or require more experience than the finite training phase. When evaluative feedback remains available after deployment, it provides a means to narrow this gap: a reward signal that reveals how well the agent performs in the situations it encounters. Not learning from this signal, and restricting agents to retraining based on human knowledge, leaves performance on the table as envisioned in Figure 2. This is the *measurable* deployment setting, where the best agents overcome their representational or computational limitations by never stopping to learn from this evaluative signal. Deploying such agents to a big world with an evaluative reward signal is a CRL problem.

Given the incompatibilities of the MDP in Section 2.1, we use the history process formalism to characterize why measurable deployment is a CRL problem. The worlds that deployed agents find themselves in change through:

1. **Action-induced Non-Stationarity**: Each history $h_t$ instantiates a new $e(h_t \cdot h, a)$ for future interactions. The agent's policy shapes the distribution of future histories it encounters. For example, a recommendation system (agent) that repeatedly suggests certain content changes the user's preferences (environment). The distribution over observations for future interactions differs from the current one precisely because of past actions. Other examples include generative models adapting to updated safety/alignment policies, or markets responding to automated trading strategies[2].

2. **Changes in Environment Dynamics**: The environ-

ment can also change due to factors outside the agent's control (seasonal variations, hardware aging, market shifts, regulatory changes). These changes can be periodic (multi-timescale patterns like diurnal or seasonal cycles) or stochastic (unpredictable shifts in dynamics).

3. **Evolving Goals**: Under the reward hypothesis, an agent's "goals and purposes" are expressed through the reward function (Sutton, 2004; Littman, 2017). This function can change over time, altering what constitutes desirable behavior even when the underlying environment dynamics remain stable. In the history process formalism, the reward function induces the agent's preference relation over histories (Bowling et al., 2023). In deployment, this preference can evolve as stakeholder priorities shift, regulations change, safety constraints are updated, or new capabilities are added to deployed systems. This challenge intensifies in multi-objective settings, where agents must balance multiple competing objectives, as both the set of objectives and their relative importance can evolve during deployment in ways unforeseen during the training phase. Unlike environmental or action-induced non-stationarity, evolving goals are often imposed by human designers or stakeholders, making it particularly relevant for real-world deployment where the goals and purposes we wish the agent to achieve are subject to change.

4. **Emergent Novelty**: New action-observation sequences can arise after deployment, lying outside the distribution of histories experienced in a finite training phase. The Big World Hypothesis formalizes why this is inevitable: the world's complexity is much richer than the representational capacity of any agent within it. Recent work on computationally-embedded agents shows how any agent embedded in its environment is implicitly constrained by its finite capacity, so there necessarily exist action-observation sequences the agent cannot realize (Lewandowski et al., 2025). A dangerous form of emergent novelty are *black swan events*: rare, highly negative rewards that agents may misperceive as impossible despite their non-zero probability (Taleb, 2008; Lee et al., 2025). Deployed agents inevitably encounter histories for which their current policy is inadequate, whether due to genuine novelty beyond its capacity or a misperception of rare events. CRL views such novelty not just as a failure mode to be detected and flagged, but as an inherent property of deployment under the Big World Hypothesis.

These four characteristics demonstrate why agents in measurable deployment should never stop learning. We now examine production systems that have successfully deployed continual RL, analyzing which challenges each faces and how continual adaptation enables their success.

---

[2]This closely relates to the literature on *performative prediction*, which studies how predictions influence their targets (Perdomo et al., 2020). This phenomenon is relevant to deployment and merits further study in the applied RL community.

### 3.1. Case Study I: Cursor Tab

Cursor Tab is a code completion system that predicts developers' next actions across their codebase, handling over 400 million requests a day (Jackson et al., 2025). The system employs online RL for their code suggestion policy, $\pi_{\theta}$, using a policy gradient method optimizing the single-step reward $J(\boldsymbol{\theta}) = \mathbb{E}_{s \sim P, a \sim \pi_{\theta}}[R(s, a)]$. New models are rolled out frequently throughout the day to get "fresh" on-policy rewards for $\widehat{\nabla_{\boldsymbol{\theta}} J}$ in the gradient ascent update. This differs starkly from most large language model providers, who train on static datasets and release new models every few months.

This deployment illustrates multiple CRL challenges. **Emergent novelty** is inherent: at 400 million daily requests across diverse codebases, the system continually encounters code patterns outside its training sets. **Changes in environment dynamics** emerge as user preferences and libraries evolve.

Cursor's use of policy gradient methods illustrates how continual learning *solutions* impose their own constraints beyond those arising from the *problem* itself. The policy gradient theorem requires on-policy samples (Sutton et al., 2000). Once the policy parameters are updated, previous user interaction data becomes stale for computing accurate gradient estimates. This is not an environmental characteristic necessitating continual learning, but rather the chosen solution's characteristic (akin to how catastrophic forgetting or plasticity loss are challenges with artificial neural networks, rather than characteristics of the problem). Nevertheless, this solution-level constraint creates operational requirements that demand tight deployment-training loops, with Cursor Tab achieving 1.5-2 hour iteration cycles between deployment and data collection for subsequent updates.

The algorithmic demands of this solution align well with the demands of the problem: the continuous deployment cycles required for on-policy learning enable both adaptation to emergent novelty and shifting environmental dynamics. This synergy demonstrates how CRL solutions are not just viable, but advantageous in deployment. Cursor Tab yielded a 21% reduction in suggestions shown while achieving a 28% higher acceptance rate (Jackson et al., 2025), exemplifying the benefit of deployed agents that never stop learning.

### 3.2. Case Study II: Lyft

Lyft's ridesharing platform processes hundreds of millions of rider-driver matches per year, requiring real-time decisions about which driver to assign to each incoming ride request (Azagirre et al., 2024). New approaches incorporate increasingly realistic features, yet typically operate with a fixed policy during deployment, with learning or calibration occurring offline or via periodic retraining (Karp et al., 1990; Bertsimas et al., 2019; Yan et al., 2020).

Lyft's 2021 deployment was the first documented case of a ridesharing matching algorithm that used online RL to learn and update its policy after deployment using real-time feedback from matching outcomes. This deployment exemplified **action-induced non-stationarity** as its primary challenge. Each matching decision directly altered the future environment. Assigning a driver to a rider relocated that driver, changed the spatial distribution of driver supply, and altered future matching possibilities.

**Environment dynamics change continually** as demand patterns vary by time of day, season, one-off events, and market conditions that evolve across cities. Out-of-distribution **novelty** can emerge from events like a popular concert, a global sporting event, or a huge conference. These persistent non-stationarities cannot be fully anticipated offline.

The choice of online RL as a *solution* addresses these *problem* characteristics directly. By continually updating value estimates from real-time matching outcomes, the system adapts to both the non-stationarity it induces through its own decisions and the external dynamics of the ride-sharing market. The authors note that "trusting an algorithm that can update itself is difficult," highlighting the operational and organizational barriers to safely deploying CRL systems. Extensive switchback experiments were therefore used to validate safety and performance prior to a global rollout.

Lyft's results demonstrate the practical benefits of continual RL in deployment. Their online RL matching system enabled drivers to complete millions of additional rides per year, generating over $30M in additional annual revenue while also improving rider pickup times and driver utilization (Azagirre et al., 2024). The authors emphasize such across-the-board improvements are rare in large-scale marketplace optimization, and attribute this to the system's ability to adapt online rather than optimize for a fixed model.

### 3.3. Case Study III: The Sim-to-Real Gap

The challenge of transferring policies trained in simulation to the real-world (referred to as the *sim-to-real* gap) provides compelling evidence for continual learning after deployment. This gap is a direct consequence of the conventional train-then-fix paradigm. Despite simulation's advantages in sample efficiency, cost, and safety for initial training in domains like robotics, policies trained purely in simulation often fail when transferred to physical robots.

This gap arises from multiple sources of distribution shift between the environment in the simulator and the environment in which the policy is deployed. Parameters such as friction, mass, and inertia are difficult to model precisely, and real robots experience wear-and-tear over time that changes their dynamics. Visual discrepancies arise in systems that rely on computer vision: differences between rendered images in simulation and real camera observations cause policies

relying on visual input to deploy poorly. Temporal characteristics such as sensor sampling rates and latencies from sensing to actuation differ between simulation and reality. The real-world also introduces sensory noise, lighting variations, and unexpected perturbations absent from simulation, which increase the sim-to-real gap.

Sim-to-real problems can manifest all four sources of non-stationarity we identified: **environmental dynamics** differ between simulation and reality; **emergent novelties** never seen in simulation appear; **environment dynamics change** from hardware degradation; and even **evolving goals** can emerge as deployment reveals misalignments between simulated task specifications and real-world objectives.

Current sim-to-real methods acknowledge that fixed policies are insufficient. Domain randomization, system identification, and domain adaptation all aim to create policies robust to the gap, but these approaches still assume the gap can be bridged during a training phase (Zhao et al., 2020). Even with sophisticated transfer techniques, policies still require real-time data for reliable performance (Akkaya et al., 2019; Horváth et al., 2022). Research in this area has led to more data-efficient training algorithms (real-world grasping in 5,000 samples instead of 580,000 (James et al., 2019)). While such 100-fold improvements in sample efficiency are valuable for reducing training costs, it does not resolve the core underlying issue: agents cease to learn after deployment. No amount of training data can allow for autonomy in the face of the big world's observation stream.

| Challenge | Cursor Tab | Lyft | S2R |
|---|---|---|---|
| Action-Induced NS | Implicit | **Primary** | Implicit |
| Env. Dynamics changes | Implicit | Present | **Primary** |
| Evolving Goals | Present | Implicit | Implicit |
| Emergent Novelty | **Primary** | Present | Present |

*Table 1.* The prevalence of CRL challenges across deployment case studies. Primary: dominant driver of continual learning; Present: clearly manifests; Implicit: likely present but not in the foreground.

## 4. Continual Learners are the Proper Choice for Measurable Deployment

Under the history process formalism, an agent is a function mapping histories to action distributions, and its *learning rule* determines how this mapping evolves. Learning can be viewed as a search over a policy set: at each history, the agent either continues searching or terminates by settling on a policy for all future interactions[3]. An agent that terminates the search is a non-continual learner. An agent that

---

[3]Note there is flexibility in what counts as "settling" on a policy. One can define an agent that stays $\epsilon$-close to some $\pi \in \Pi$ as having stopped its search. This is relevant for real-world deployment, where there are acceptable ranges of competent performance.

continues searching indefinitely, adapting its policy with experience, is a continual learner.

Abel et al. (2023) formalize this distinction through the notion of a *best agent*: one that maximizes a specified performance measure over the set of policies reachable by its learning rule. A problem is an instance of CRL if and only if the best agent never terminates its search over the policy set. Crucially, this definition depends on the choice of policy set and learning rule. Changing either can switch a problem or a learner from continual to non-continual, or vice-versa.

**A Minimal Example.** Consider a parameterized policy, $\pi_\theta$, implemented as a single-layer, fully-connected, artificial neural network, with 32 units and a scalar input and output. We can consider the space of possible parameter configurations $\theta \in \mathbb{R}^{64}$ as our policy set. The learning rule would be the optimization algorithm we choose for switching between these parameter configurations, such as Stochastic Gradient Descent (SGD) with an annealing step-size. If the step-size is set to zero eventually, the learning rule settles on one specific $\theta_{\text{final}}$. From this point, the agent stops searching and commits to using that policy $\pi_{\theta_{\text{final}}}$ for all future actions, making it a non-continual learner. This is fine if a $\pi^\star$ truly does exist, and the policy set is rich enough to contain $\pi^\star$ within a reachable amount of compute, but that is often not the case in real-world deployment. If instead we meta-learn the step-size for SGD with IDBD (Sutton, 1992), the step-size never converges to zero, and the agent never settles on a $\theta$. This agent qualifies as a continual learner.

**Measurable Deployment through the Lens of CRL Agents.** The question for any deployment is first determining whether the optimal policy $\pi^\star$ is practically reachable within the agent's policy set under available resources.

If yes, as is the case with the checkers engine, the agent can terminate its search upon finding $\pi^\star$. The problem is not an instance of CRL as the best learner can stop its search.

If no (as we argue is the case for most real-world deployments), then no practically reachable element of the policy set maximizes performance across all reachable histories, and the problem is a CRL problem. In measurable deployment, the agent still receives an evaluative signal revealing how well it performs. The best agents leverage this feedback through their learning rule to update their policy, improving performance over time and continuing to adapt as new histories are encountered. This is precisely the CRL setting: the best agent cannot terminate its search, because no reachable policy is optimal. The only path to better performance is to keep adapting, leveraging evaluative signal.

**Two Learning Rules Compared.** Figure 2 illustrates two responses to measurable deployment. The left plot shows a single train-then-fix cycle: an agent searches for a high-performing element of the policy set and pauses the search

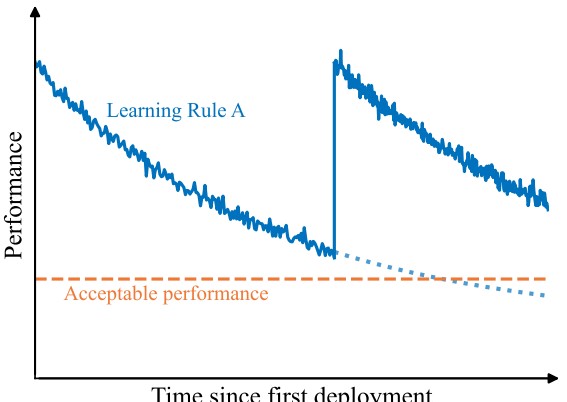 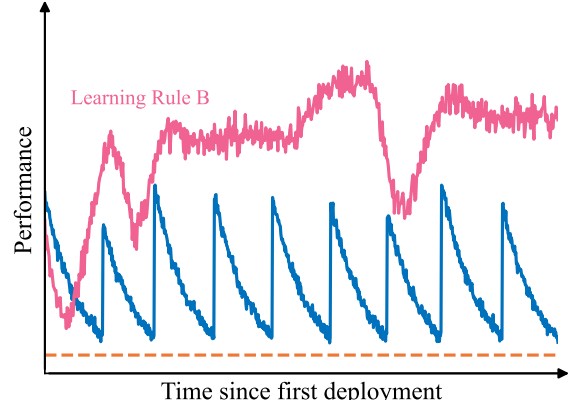

*Figure 2.* The train-then-fix paradigm (Learning Rule A) versus our vision for a continual learner (Learning Rule B) in measurable deployment. **Left:** A fixed policy degrades until retraining is triggered (orange dashed line marks acceptable performance, dictated by human expertise). **Right:** Two learning rules compared over extended deployment: periodic retraining (blue, sawtooth) versus a continual learner that decides when and how to update its policy, without human intervention (pink).

upon finding one, settling on a fixed policy. Due to the non-stationarities identified in Section 3, performance degrades over time until a minimum performance threshold set by human experts triggers another round of search for a high-performing policy. The right plot extends this over a longer deployment horizon. This leads to a sawtooth pattern from repeated retraining. Learning rule A treats deployment as a series of non-continual problems. The Rusting Pendulum environment in Appendix A empirically demonstrates this degradation: a train-then-fix policy fails as joint friction accumulates, but a continual learner maintains performance.

In contrast, the agent using learning rule B never stops its search. Recognizing that $\pi^\star$ is not in its policy set, the agent continually adapts, using the available evaluative signal to guide its ongoing search.

Both A and B are learning rules (maps from a history to a distribution over policies), but only the latter embraces the CRL problem setting. They differ in whether the agent designer accepts measurable deployment as inherently a CRL problem.

We argue that designers facing measurable deployment should embrace the latter framing. Abel et al. (2023) demonstrate this empirically in a controlled switching MDP where a continual $Q$-learner with a fixed step-size consistently outperformed one that converges. Sutton et al. (2007) showed that tracking outperforms convergence even in a stationary environment when observations are non-Markovian, supporting the case for continual adaptation is broader than just non-stationary settings. By accepting the continual nature of the problem and designing agents that never terminate their search, we enable them to leverage the evaluative feedback they receive, rather than forgoing performance between retraining cycles.

## 5. Call to Action

Recognizing measurable deployment as a CRL problem is only a first step. Translating this insight into practice requires coordinated effort from practitioners and researchers. This section offers concrete recommendations for each group.

### 5.1. Recommendations for Practitioners

If your deployed system receives evaluative feedback, and operates in an environment too complex to fully characterize offline, then you are in measurable deployment. The question is not whether to adapt, but how.

**Recognize your deployment regime.** Before investing in continual learning infrastructure, assess whether your setting warrants it. Ask: (1) Does my system receive ongoing evaluative signals after deployment? (2) Does performance degrade without intervention? (3) Is periodic retraining costly, slow, or insufficient? If all three hold, you are giving up performance by fixing your policy after training.

**Build feedback loops, not just pipelines.** Traditional MLOps treats deployment as the end of learning: data flows in, a model is trained, and the artifact is served until staleness triggers retraining, leading to the sawtooth pattern in Figure 2. Continual deployment inverts this: the deployed model is the learning system, and production data is training data. This requires infrastructure changes: logging interactions in formats amenable to online updates, maintaining low-latency paths from feedback to model parameters, and treating model checkpoints as evolving rather than versioned artifacts. Cursor Tab's 1.5–2 hour iteration cycles exemplify this tight coupling between deployment and learning.

**Validate continually, not just before deployment.** Lyft's switchback experiments (alternating between policies in production to measure causal effects) are a safety validation extending beyond pre-deployment. When policies update continually, so must evaluation. Implement online monitoring for performance metrics, distributional drift, and safety constraints. Maintain fallback policies that can be activated if the learner behaves unexpectedly. The goal is not to eliminate risk, but to make adaptation safer than stagnation.

**Introduce non-stationarity deliberately.** A practical way to stress-test your system's adaptability is with controlled non-stationarities in development: perturb rewards, skew observation, or simulate concept drift. This reveals brittleness before deployment and calibrates how aggressively your system should adapt. If your system cannot handle synthetic non-stationarity, it will struggle with the real world.

### 5.2. Recommendations for Researchers

A number of prior works have surveyed the challenges faced by continual learners and outlined promising directions for future research (Parisi et al., 2019; Delange et al., 2021; Khetarpal et al., 2022; Verwimp et al., 2024). Notably, the *Alberta Plan* proposed by Sutton et al. (2022) presents a long-term research plan aimed at developing agents capable of sustained, open-ended learning. In addition, several technical studies have examined the challenges associated with deploying continual learners in the real-world environments (Amodei et al., 2016; Dulac-Arnold et al., 2019; Hamadanian et al., 2022). This section focuses on a subset of challenges and recommendations that have been highlighted more recently, with the goal of drawing attention to issues that may otherwise be overlooked and encouraging further investigation by the research community.

**Study the performance of conventional methods implemented in continual settings.** A natural recommendation is to reassess the behavior of widely adopted conventional methods when deployed in continual learning problems. Many algorithms and optimization techniques that perform well in stationary settings may exhibit undesirable behavior under non-stationarity. Degris et al. (2024) showed that two commonly used optimizers, RMSProp (Hinton et al., 2012) and Adam (Kingma & Ba, 2015), were unsuitable for step-size adaptation in a simple continual learning scenario. Despite this, these optimizers are frequently employed in continual and online settings (Han et al., 2022). CRL research should treat hyperparameter tuning as part of online deployment, not a hidden offline phase (Hakhverdyan, 2024; Patterson et al., 2024), and evaluate methods that adapt hyperparameters during learning. Notable candidates for this include Bayesian (Parker-Holder et al., 2022) and meta-gradient methods (Sutton, 1992; Degris et al., 2024).

**Rederive algorithm properties for history processes.** The shift from the MDP formalism to the history process is not merely notational. Standard convergence results for value-based and policy gradient methods depend critically on the Markov property. Without it, value functions defined on state representations are approximations of the true history-conditioned values, and their theoretical guarantees dissolve (Abel et al., 2024; Elelimy et al., 2025). This is an under-explored consequence of adopting more realistic CRL foundations. Researchers should audit which properties of conventional algorithms survive the loss of Markovianity, and work to establish what weaker guarantees can be recovered (convergence in a non-stationary sense, regret bounds under history dependence, or stability conditions for online hyperparameter methods).

**Consider different evaluation metrics.** Many CRL works evaluate agents using the expected average reward as the performance measure. For this metric to be well-defined and meaningful, the underlying environment should be at least weakly communicating, ensuring that long-run averages are independent of the initial state (Wan & Sutton, 2022). Many real-world and production systems violate this assumption due to irreversible failures, shutdowns, or terminal user states that effectively partition the state space. As a result, average-reward-based evaluation may obscure meaningful differences between agents. We recommend that researchers consider more general performance measures that reduce implicit assumptions about environment structure and better reflect effective performance in continual deployment settings. One such evaluation metric is *deviation regret* (Elelimy et al., 2025), which assesses agents based on the situations they encounter rather than solely on asymptotic reward accumulation.

**Take resource constraints into account.** Finally, CRL research should explicitly account for finite computational and memory resources. Under the Big World Hypothesis (Section 3), environmental complexity exceeds agent capacity, shifting the objective from convergence to tracking targets over time (Sutton et al., 2007). Recent theoretical results formalize this intuition, showing that continual learning is necessary for resource-constrained agents to sustain performance (Kumar et al., 2025). Practically, this motivates architectures that dynamically reallocate computation and memory, favoring efficient, approximate updates for real-time adaptation (Javed et al., 2023; Lo et al., 2024; Vasan et al., 2024; Tamborski & Abel, 2025).

### 5.3. Benchmarks

Benchmarks have always played an important role in fostering collaboration and advances in machine learning (Martínez-Plumed et al., 2021). From CIFAR (Krizhevsky,

2009) and ImageNet (Krizhevsky et al., 2012), that challenged and accelerated computer vision, to mathematics benchmarks that have been critical in the progress of reasoning in large language models (Fang et al., 2025).

The lack of such benchmarks in CRL has been one of the main barriers in research in this field (Khetarpal et al., 2022). The current CRL benchmarks add non-stationarities to existing RL benchmarks. This may be by changing the dynamics of the environment, switching between different tasks (Switching Arcade Learning Environment; Abbas et al., 2023), or changing the reward function over time (Anand & Precup, 2023). There are also benchmarks inspired by robotics (Wolczyk et al., 2021), and games that introduce non-stationarities gradually (Mohamed et al., 2026).

We suggest that benchmarks must be developed based on defined characteristics of CRL problems to be useful for the community. In particular, building on the proposed need for a big world simulator (Kumar et al., 2024) for continual learning, where there are no diminishing returns to increasing the capacity of the agent, as any agent with a finite capacity will need to learn and update forever in such an environment. This is in line with deployment settings.

## 6. Alternative Views

To stimulate productive discussion in the community, we present several alternative views to our thesis that deployed RL should be continual. While these concerns merit consideration, we argue they can be addressed through careful system design, deployment best practices, and by reorienting research toward continual learning rather than convergence to fixed artifacts. Much of our community's research remains on algorithms that solve problems and stop learning, which leads to deployments lacking continual adaptation.

**Current approaches are already adaptive.** One can argue that deployed systems already adapt continually through modern MLOps pipelines. Periodic retraining and fine-tuning are legitimate learning rules (under Section 2's definition). Both curves in Figure 2 are valid continual learning rules: they differ not in whether adaptation occurs, but in how it is triggered. When retraining is integrated as an internal, autonomously triggered rule, it can constitute genuine continual adaptation in the CRL setting.

However, retraining is often part of an external MLOps pipeline in the form of human intervention/expertise, rather than the agent's internal operation. This limits how and when the agent updates its policy to human knowledge, rather than letting the agent discover a good time and direction for an update.

In contrast, when re-training is integrated as an internal learning rule within the agent's autonomous operation, the agent gets to decide how and when it updates its policy based on what it has seen. In domains that are understudied or unexplored by humans (exploring extra-terrestrial environments), or where optimal performance is unbounded (stock trading), such external updates may not adequately address the demands of measurable deployment.

**Policies trained offline are generalizable enough.** Zero-shot and few-shot generalization, and sim-to-real methods may provide favorable initializations at deployment by distilling prior knowledge so competent performance is reached faster (Kirk et al., 2023; Beck et al., 2023; Iannotta et al., 2025). These are worthwhile complements to continual learning from interaction, but not substitutes. They accelerate early adaptation but are bounded by the dynamics seen during a finite training phase. As the deployment horizon extends and the agent encounters the non-stationarities from Section 3, the best agents must continue adapting beyond what any prior can cover. In-context approaches (Brown et al., 2020) can in principle be viewed as a continual learning rule by treating context as part of a recurrent hidden state (Akyürek et al., 2023; Kang et al., 2025), but whether finite context windows support credit assignment and exploration under measurable deployment remains an open question.

**Not all deployments need continual learning.** While not all deployments require continual learning, we argue that continual learning is a requirement for *measurable* deployment. As discussed in Section 3, real-world deployments are often in the Big World regime and are subject to the four non-stationarities from a variety of causes (ranging from gradual wear and tear to sudden black swan events) that cannot be fully anticipated by designers or represented within the agent. In such a setting, the agent should leverage reward as evaluative feedback to adapt continually.

For example, an industrial robot arm repeating the same task seems stationary, but mechanical wear and joint degradation accumulate, causing accuracy to drift. What appears stationary is revealed as non-stationary over extended deployment.

**Reward signals may be sparse, delayed, or unobservable after deployment.** Our argument hinges on measurable deployment, but what if the evaluative signal cannot be obtained reliably? A Roomba in a factory has extensive instrumentation to test and reward all scenarios; in a home, feedback is sporadic at best. A sensor that degrades may report misleading observations, poisoning any learning signal.

When rewards are delayed beyond the horizon of useful credit assignment, an agent may still be able to learn from the differences in values of the situations it encountered. When reward is absent entirely, there is nothing to learn from. Monitored-MDPs formalize this challenge, and agents train reward models to still learn with partial feedback

(Parisi et al., 2024; Mohammedalamen & Bowling, 2026). Such monitors are an open problem in history processes, which do not have the same convergence guarantees as a Markovian monitor, and we acknowledge this limitation. Our position applies specifically to measurable deployment, where an evaluative signal exists and can be trusted. The concerning observation is that many deployed systems have access to reliable feedback (click-through rates, revenue, suggestion acceptance rates), but still default to fixed policies. We urge the community to reconsider such cases.

**Safety and alignment concerns favor fixed policies over continual learners.** One might argue that deploying fixed policies that have been extensively validated offline is inherently safer than allowing agents to continue learning after deployment. A fixed policy's behavior can be exhaustively tested, its failure modes catalogued, and its performance bounds characterized before it is deployed. However, fixed policies are not a viably safer alternative. As established in Section 3, the four sources of non-stationarity ensure models cannot remain static. Fixed policies either degrade unsafely or require human intervention, both carrying their own safety risks. The wealth of safe RL literature reflects that adaptation is necessary for maintaining safety, not antithetical to it (Moldovan & Abbeel, 2012; Alshiekh et al., 2018; Kumar et al., 2020; Thomas et al., 2021; Skalse et al., 2022; Moghimi & Ku, 2025). Recent work has even shown that agents can be trained to behave cautiously in the face of unseen observations (Mohammedalamen et al., 2021).

Safe AI is a field still in its infancy and new safety constraints and legislation are frequently announced and updated. Instead of re-training new models from scratch to adhere to them, continual learning may allow a learner to adapt to these regulatory updates.

## 7. Conclusion

We have presented why deploying decision-making agents that are incapable of optimality, but privileged with evaluative feedback in the form of a reward signal after deployment, is a continual reinforcement learning problem. To largely ignore this signal, and restrict the agent to rely on human intervention or expertise to decide when and how the policy is updated, is to leave performance on the table indefinitely. We hope this paper encourages practitioners to leverage the feedback their systems already receive, and researchers to prioritize continual learning as the problem setting for deployed RL.

## Acknowledgements

The authors would like to thank Michael Bowling, A. Rupam Mahmood, Khurram Javed, Gautham Vasan, Montaser Mohammedalamen, Diego Gomez, Abhishek Naik, Andrew Patterson, Shibhansh Dohare, and Matthew Schlegel for constructive discussions and feedback on an earlier draft of this work. We thank the anonymous reviewers and area chair for their time, reviews, and suggestions that strengthened this paper. Portions of this research is based on work supported by the Canadian AI Safety Institute Research Program at CIFAR, funded by the Government of Canada.

## Impact Statement

This paper presents work whose goal is to advance the field of Machine Learning. There are many potential societal consequences of our work, none which we feel must be specifically highlighted here.

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

# A. Rusting Pendulum

We will use a toy problem with wear-and-tear to demonstrate the advantage of continual adaptation over train-then-fix.

To do so, we introduce the **Rusting Pendulum** environment, a variant of the original Pendulum classic control environment. As in the original Pendulum, the agent's goal is to raise a rigid, uniform rod that is hinged on one end, from the downward to the upright position by applying continuous-valued torques as actions. At each timestep $t$, the agent observes the $x_t$ and $y_t$ coordinates of the rod along with its angular velocity $\dot{\theta}_t$. The reward, $r_{t+1}$, is negative at each timestep, and is closer to $0$ the more upright and stable the pendulum is. We introduce a non-stationarity by modifying the environment's transition dynamics (Towers et al., 2024) to include a damping term, $-b_t\dot{\theta}_t$. The damping coefficient $b_t$ grows in a noisily quadratic manner over time, as shown in Figure 3 (Gaussian noise $\sigma = 0.02$). This mimics rust that accumulates at the pendulum joint, making actions from older policies lack the necessary torque to overcome the rust and raise the pendulum.

$$\dot{\theta}_{t+1} = \dot{\theta}_t + \left(\frac{3g}{2l}y_t + \frac{3}{ml^2}A_t - b_t\dot{\theta}_t\right)\Delta t,$$

$$x_{t+1} = \cos(\theta_t + \dot{\theta}_{t+1}\Delta t),$$

$$y_{t+1} = \sin(\theta_t + \dot{\theta}_{t+1}\Delta t),$$

$$r_{t+1} = -\left(\theta_t^2 + 0.1\,\dot{\theta}_t^2 + 0.001\,A_t^2\right),$$

where $A_t \sim \pi_t$ is the torque applied at timestep $t$, and $\Delta t$ the timestep duration.

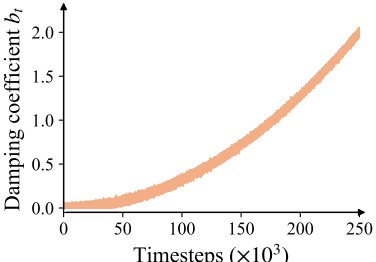

*Figure 3.* Growth of the damping coefficient over the experiment.

Figure 4 (left) shows two agents in the Pendulum environment. The solid curve is a Sarsa($\lambda$) agent (Rummery & Niranjan, 1994), continually updating its policy[4]. The dashed curve is the train-then-fix paradigm, where checkpoints of the solid curve's policy are taken every 30k steps and ran without updates. Both agents run for a single, 250k step episode, with no resets. The colour of the dashed curve maps to the time of checkpointing from the solid curve. In Pendulum, the train-then-fix policy matches the continual learner's performance as later checkpoints learn to recover from unfavorable states. In Rusting Pendulum (Figure 4, right), the dynamics shift faster than the checkpoints account for. A policy trained at low friction has not learned the larger torques to recover under rustier dynamics, and its performance degrades. The same trend is observed across runs (Figure 5). Spending compute tuning the learning rate may reduce this gap on a problem-to-problem basis.

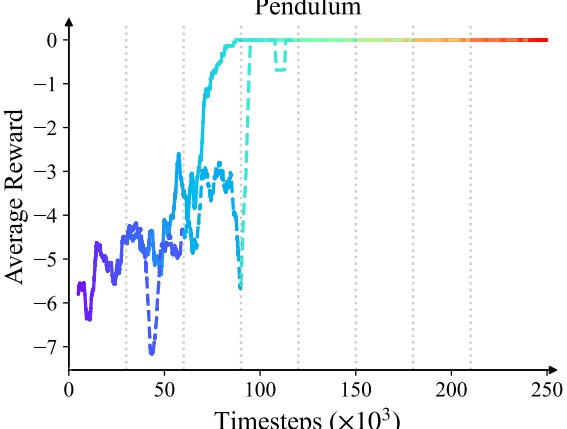
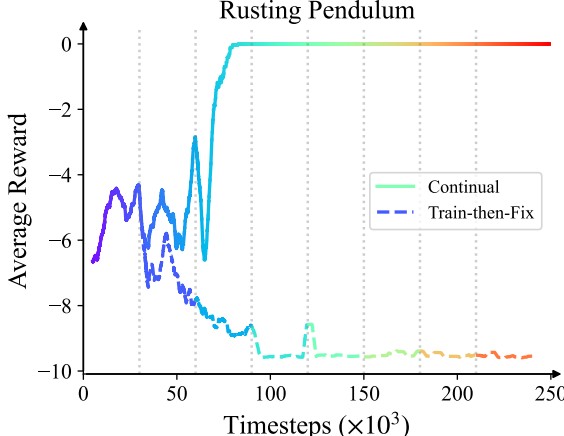

*Figure 4.* Continual learning vs train-then-fix on two pendula. **Left:** In Pendulum, the train-then-fix policy (dashed) matches the continual learner after convergence, as the optimal policy does not change. **Right:** In Rusting Pendulum, the train-then-fix policy degrades as the dynamics drift away from those seen during training, while the continual learner adapts and maintains near-optimal performance. Colour encodes time progression. Each dashed vertical line corresponds to a checkpoint taken every 30k steps.

---

[4]The value function is linear in tile-coded features (Sutton & Barto, 2018), and the action space is discretized into 7 equal bins. Both agents use a fixed step-size of $\alpha = 0.1$ and a trace decay parameter of $\lambda = 0.95$.

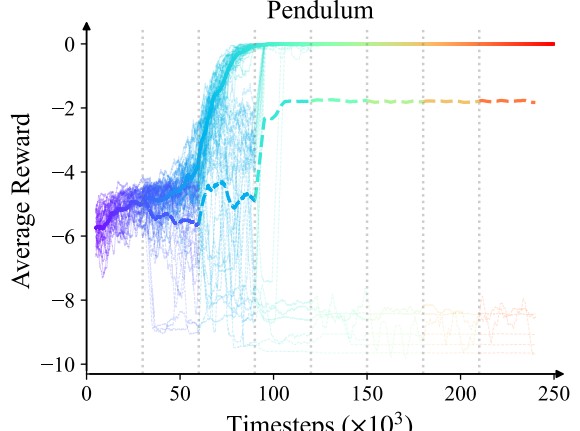 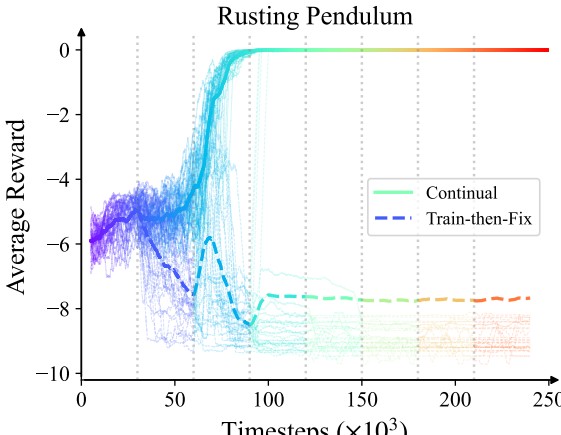

*Figure 5.* The experiment from Figure 4 repeated across 30 independent runs. Faint curves show individual runs, whereas the opaque curve is the mean performance. Confidence intervals did not accurately capture the distribution of failure modes.

