# OpenReview forum: "Position: Deployed Reinforcement Learning should be Continual"
_ICML.cc/2026/Position_Paper_Track — ICML 2026 Position Paper Track regular_

### Official Review · Reviewer_3myq · 2026-03-10

**Significance:** 3
**Argument Clarity:** 4
**Rating:** 5
**Confidence:** 4

**Questions:**

* Do you believe that meta-RL and zero-/few-shot generalization methods are not capable enough to achieve adaptability at test time, when compared to continual RL? (relevant references are given in the Strengths & Weaknesses Section)

**Alternative Views Section:**

Yes

**Compliance With Llm Reviewing Policy A Conservative:**

Affirmed.

**Discussion Potential:**

4

**Final Justification:**

See the acknowledgement

**Paper Summary:**

The work argues that reinforcement learning agents could make use of an evaluation metric at deployment time, then this should be leveraged. Thus, the work advocates for continual learning solutions rather than simply deploying fixed policies. The work discusses what is deemed "measurable deployment" and why this ultimately leads to a continual reinforcement learning problem. The work further evaluates three case studies to highlight how continual learning (even in a simple form of periodic retraining) has achieved impressive real-word results. Based on this, the work presents a call to action, including recommendations for practitioners and researchers alike on how to improve the state of deployable RL. Finally, the work discusses alternate views that are opposed to the stated position.

**Position:**

Yes

**Position In Title:**

Yes

**Related Work:**

3

**Strengths And Weaknesses:**

The work is well written and clearly argues for the position. As the work is concerned with deployable RL, I particularly appreciate the discussion of case studies that highlight how real world deployment is evaluated.

I further believe that Section 5 (Call to Action) provides a lot of utility to the community and the recommendations should be highlighted already much earlier in the text or even the abstract. I believe that this could draw in more readers that aim to build deployable RL.

The biggest weakness in my opinion is the alternative views section. In my opinion meta-learning (https://arxiv.org/html/2301.08028v3 or https://openreview.net/forum?id=UENQuayzr1) and zero-/few-shot generalization (https://arxiv.org/abs/2111.09794 or https://arxiv.org/abs/2202.04500) approaches should be discussed as they aim to learn policies that are adaptable with out (or as little as possible) retraining. I believe that the argument here would be that with such approaches aim to provide more system/environment knowledge to a policy such that it can detect how to best adapt at test time. Particularly with Case-study 3 in mind it has been shown that such methods can bridge the sim2real gap (https://arxiv.org/abs/2511.04249).

Beyond this criticism I believe the paper will lead to interesting discussions and am in favor of acceptance.

**Support:**

3

---

> ### Author Rebuttal · Authors · 2026-03-30
>
> We thank reviewer 3myq for their time and effort thoroughly reviewing this manuscript, and are glad to hear their appreciation for the case studies and call to action. We are happy to alert the readers of the actionables earlier in the paper.
>
> **Q1)**\
> We thank the reviewer for raising these alternatives. We agree that meta-learning is a natural complement to continual RL rather than a competing paradigm: methods that adapt hyperparameters or optimization strategies online can be understood as instantiating a never-ending policy search more efficiently, which aligns directly with our position. We will incorporate the meta-learning references [1,2] into our second recommendation for researchers “Study the performance of conventional methods when implemented in continual settings” (Section 5, line 352), if the paper is accepted.
>
> Zero- and few-shot generalization methods occupy a slightly different role: we view them as providing a favorable initialization at deployment time, distilling prior human knowledge so that competent performance is reached faster. This is valuable, particularly early in deployment, but these approaches are ultimately bounded by what can be anticipated during a finite training phase. As we note in the paper, limiting a learning system to human knowledge is at odds with the Big World Hypothesis, and deployment will inevitably surface conditions that fall outside any finite prior. That said, few-shot generalization can greatly accelerate early adaptation, as the reviewer pointed out [3,4,5], but the best agents will still need to keep learning as the deployment horizon extends. This makes few-shot and zero-shot a worthwhile complement to continual learning, but not a substitute. We recognize this as a valid point worth explicitly mentioning in the manuscript, if accepted.
>
> Please do let us know if there are any other nuances to this we did not account for.
>
> [1] Beck et al. 2023, A Survey of Meta-Reinforcement Learning\
> [2] Shala et al. 2025, Efficient Cross-Episode Meta-RL\
> [3] Kirk et al. 2023, A Survey of Zero-shot Generalisation in Deep Reinforcement Learning\
> [4] Benjamins et al. 2023,​​Contextualize Me -- The Case for Context in Reinforcement Learning\
> [5] Iannotta et al. 2025, Can Context Bridge the Reality Gap? Sim-to-Real Transfer of Context-Aware Policies

---

> > ### Author Rebuttal · Reviewer_3myq · 2026-04-01
> >
> > Thank you very much for the response. I was already very positive about the paper and keep my score.
> >
> > As the other reviewers all also asked for different aspects of generalizable RL solutions or generalizability at large, I am wondering why the authors deem the "Big World Hypothesis" to be so pertinent for this position. An alternative view that might be worth discussing is the view of many (small) worlds (see e.g. https://rlj.cs.umass.edu/2024/papers/Paper167.html). Particularly, from my understanding of continual (reinforcement) learning, big distribution shifts could be seen as starting in a novel small world. As knowledge transfer is quite possible across such related worlds, a broader deployment horizon might not necessitate continual learning as such. However, I do agree that there will always be a region out-of-distribution that will make such approaches fail, while continual learning continues to excel.

---

### Official Review · Reviewer_zspP · 2026-03-13

**Significance:** 3
**Argument Clarity:** 3
**Rating:** 5
**Confidence:** 4

**Questions:**

Questions:
1. Under the history process formalism used in this paper, are there any unstated assumptions about the degree of non-stationarity in the environment? In particular, if we assume that the transition function can change arbitrarily between timesteps, it seems like nothing is learnable since no prior experience can help inform future decisions.
2. Where would a meta-learned in-context RL agent fit into the framing of this paper? The resulting model is technically a static history-conditioned policy (with frozen weights) but it can also be framed as a continually learning agent where the policy is parameterized by the hidden state and the learning rule is defined by the model.

**Alternative Views Section:**

Yes

**Compliance With Llm Reviewing Policy A Conservative:**

Affirmed.

**Discussion Potential:**

4

**Final Justification:**

See acknowledgement. Overall, still positive and supportive of acceptance but want to clarify some details.

**Paper Summary:**

Modern RL systems typically have a separation between training and deployment phases, which can lead to performance degradation and a need to re-train after a period of deployment. This position paper argues that these systems should be framed as a continual learning problem when reward signals are available, blurring the line between training and deployment. This work analyzes three case studies to reveal the prevalence of non-stationarity in deployed RL and provides actionable advice to develop practical continual learning methods.

**Position:**

Yes

**Position In Title:**

Yes

**Related Work:**

4

**Strengths And Weaknesses:**

Strengths:
- The setting is clearly laid out and preemptively addresses potential misunderstandings (like the distinction between continual learning problems and solutions)
- The case studies provide a clear narrative to defend the position of the paper with useful examples.
  - I found the note about hardware degradation for robotics as a source of non-stationarity particularly interesting. This is rarely discussed within RL robotics papers, where deployment is typically just short demos, but this is an important component for long running robots.
- The call to action gives advice that is consistent with the thesis of this paper


Weaknesses:
- The “minimal example” reads a bit awkwardly due to the arbitrary details of the neural network architecture; it feels like a set up to an experiment section that doesn’t exist.
- Minor point: Under the history process formalism, it is a bit strange to have the policy condition on state instead of action-observation history, given that the state is typically only partially observed.

**Support:**

3

---

> ### Author Rebuttal · Authors · 2026-03-30
>
> We thank the reviewer zspP for their positive evaluation of the paper and are glad to see their interest in its case studies and problem setting. We hope the following responses adequately address the questions, and are more than happy to elaborate if not.
>
> **Q1)**\
> We appreciate the reviewer raising this point as it gives us an opportunity to clarify an important distinction. The reviewer is correct that the history process formalism makes no assumptions about the environment, including whether a learnable goal exists. This generality is intentional and is precisely what makes the formalism a powerful and flexible modeling tool.
>
> That said, our claim about deployment is not about what the formalism assumes, but about what deployment itself implies. The formalism can model environments where nothing is learnable, and we do not dispute that. The key point is that practitioners would not choose to deploy a reinforcement learning system in such environments to begin with. A fully arbitrary environment where no prior experience can inform future decisions is not a setting where any learning approach would yield meaningful results. This is a property of the environment itself, not a limitation of the formalism.
>
> In practice, this degenerate case rarely arises in deployment settings. When stakeholders decide to deploy an RL system, they do so because they believe there is a desirable behaviour or goal the system can meaningfully learn to pursue. The history process formalism does not rule this out but simply does not require it, which is the appropriate stance for a general-purpose formalism. We believe that the generality of the formalism and the practical preconditions of deployment are therefore not in conflict.
>
> **Q2)**\
> We thank the reviewer for this thoughtful framing and acknowledge that the latter interpretation has merit since the agent's behavior changes as the context grows, without any explicit training phase.
>
> Under the framing of this paper, a meta-learned in-context RL agent falls closer to the fixed interpretation. Following [1], learning is an ongoing search process that never terminates. In a meta-learned in-context RL agent, that search process terminates at the end of the meta-training phase and the weights, which encode the learning algorithm itself, are frozen. What happens at deployment is reasoning conditioned on context, not long-term learning. A simple test illustrates this: if the context window were cleared mid-deployment, the agent would lose all accumulated knowledge. It is therefore difficult to consider context as a policy parameterization for continual learning. A genuinely continual agent would carry that knowledge forward in its weights, even without any prior context.
>
> We do agree such an agent could be sufficient if all possible dynamics could be encapsulated within its context, but under the Big World Hypothesis this is rarely satisfied in practice [2]. Consider an actuator failure where the agent must discover alternative action combinations to achieve its goal. If this substitution pattern was seen during meta-training, an in-context agent could potentially handle it. If not, the agent faces two fundamental problems: it has no principled mechanism for exploring new action combinations, and even if it stumbled onto the correct alternative, accurately assigning credit through a context dominated by failed attempts would be extremely difficult. What the in-context agent lacks is the ability to genuinely discover new solutions and durably encode them, something only weight updates can provide. Continual learning is not an alternative to in-context adaptation; it is the mechanism that handles what in-context adaptation fundamentally cannot. We recognize this as a valid point worth explicitly mentioning in the manuscript, if accepted.
>
>
> [1] Abel et al. 2023, A definition of Continual Reinforcement Learning\
> [2] Javed and Sutton 2024, The Big World Hypothesis

---

> > ### Author Rebuttal · Reviewer_zspP · 2026-04-01
> >
> > Thanks for the responses! I will maintain my score, but I want to push back on some points made in the rebuttal.
> >
> > I agree that the formalism is "flexible" but I don't believe it is "powerful" in the sense that it is immediately useful for developing and analyzing algorithms. As an example (separate from continual learning), the classic MDP formulation of RL directly motivates the derivation and design of many RL algorithms because of its (relatively) strong assumptions. The history process lacks useful structure in itself; the utility comes from the deployment setting itself, as mentioned in the rebuttal. It seems like revisiting the formalization should be an important component for the development of continual learning as a field.
> >
> > The meta-learning discussion is nuanced and I appreciate the points raised to 3myq and me in the rebuttals. I disagree with the claim that in-context adaptation cannot fundamentally do continual learning. In particular, any continual RL rule can be encoded into a neural network, where the typical policy "weights" are instead the hidden state, meaning in-context RL is strictly more general (note that I am not assuming the transformer architecture but instead allow a more complex recurrent architecture). The "simple test" arguing that knowledge needs to be stored in weights seems somewhat arbitrary; it is analogous to arguing that humans don't learn because our DNA doesn't change over our lifetime and a complete brain transplant would cause us to lose our memories. Following the definition of continual learning as a neverending search process, we can re-contextualize in-context learning as searching for the best policy parameterized by the recurrent states (instead of network weights), following the search algorithm defined by a learnt algorithm (instead of a hard-coded RL algorithm).

---

### Official Review · Reviewer_C5PX · 2026-03-13

**Significance:** 3
**Argument Clarity:** 2
**Rating:** 4
**Confidence:** 4

**Questions:**

1. Could the authors define measurable deployment more clearly, and specify which kinds of feedback signals fall within this category and which do not?

2. How does the problem formulated in this paper relate to RL generalization, the deployment gap, and test-time adaptation? Could the authors’ perspective be understood as an alternative framing of these existing problems?

3. For cases such as Cursor, could the authors provide a more concrete description of how the system is implemented, or add a small-scale experiment to more directly support the paper’s central claim?

**Alternative Views Section:**

Yes

**Compliance With Llm Reviewing Policy A Conservative:**

Affirmed.

**Discussion Potential:**

3

**Final Justification:**

See the acknowledgement.

**Paper Summary:**

This paper argues that reinforcement learning systems that have already been deployed and can still receive some form of evaluative feedback after deployment should be treated as a continual reinforcement learning problem, rather than remaining within the paradigm of “train once, then deploy a fixed system.” The authors contend that real-world deployment environments are typically non-stationary, so systems need to keep adapting after deployment. Drawing on several case studies, the article seeks to show that deployed RL should be understood and studied more from the perspective of continual learning.

**Position:**

Yes

**Position In Title:**

Yes

**Related Work:**

2

**Strengths And Weaknesses:**

Strengths:
This position paper addresses a topic that the community should seriously discuss. Whether post-deployment RL systems should be viewed as a continual learning problem is a perspective with clear practical relevance, and it also connects to ongoing discussions in the community around online adaptation, continual RL, and real-world RL. The paper is generally well organized, and its central stance is fairly clear, which gives it meaningful value as a discussion piece.

Weaknesses:

The notion of “measurable deployment” is still somewhat broad. The authors emphasize that if evaluative signals remain available after deployment, the problem should be viewed through the lens of continual RL. However, it is still not clear what kinds of feedback signals actually qualify as “measurable deployment” in this context, and the boundaries remain vague. Different types of signals vary substantially in availability, noise level, delay, and causal interpretability, so the current definition is not yet sufficiently operational.

I am not entirely sure how the problem framed in this paper relates to existing work on the training–deployment gap and on RL generalization/adaptation. To some extent, the issue the authors describe could also be understood as another formulation of the mismatch between the training distribution and the deployment distribution. In particular, directions such as RL generalization, non-stationary RL, and test-time adaptation all seem closely related to the paper’s main claim, yet the paper does not sufficiently discuss the connections and distinctions. If the authors want to argue that this is a more appropriate new framing, they need to explain more clearly how it differs from existing problem settings.

The case studies presented in the paper are suggestive, but some of the supporting arguments still feel relatively weak. For example, in the Cursor case, there does not yet seem to be enough concrete information about how the actual system implements continual learning or closes the feedback loop in practice, so it reads more like an illustrative anecdote than strong evidence. The paper would be more convincing if the authors could provide a more concrete case study, or include a small experiment showing the benefits of continual adaptation under measurable deployment.

**Support:**

2

---

> ### Author Rebuttal · Authors · 2026-03-30
>
> We are very grateful to reviewer C5PX for their perceptive review, and appreciate the points their questions raised. We address them as follows, please do not hesitate to let us know if we can clarify or remove ambiguities from any of them.
>
> **Q1)**\
> We thank the reviewer for this question, and agree that further details could help clarify our deployment setting. Measurable deployment requires two conditions: the agent is incapable of optimality (due to representational or resource constraints), and an evaluative signal remains available throughout deployment (one that is scalar, informative about the agent's goal, trustworthy, and reliable).
>
> The desiderata for what makes a good evaluative signal closely mirror those studied in the reward design literature [1,2,3,4,5,6], and referencing them would greatly help situate our measurable deployment setting. Importantly, our argument is not that such signals are universally present: reward signals that are unobservable or extremely sparse can place a deployment outside the measurable regime. Rather, our position critiques the concerning trend of many real-world systems with access to reliable evaluative feedback that meet the criteria from the literature (like click-through rates, ride-completion outcomes, code-suggestion acceptance rates) still defaulting to fixed policies over adaptation. Measurable deployment is precisely this underexploited setting, and we are very open to sharpening this concern in the writing and relating it to the reward design literature as appropriate, if the paper is accepted.
>
> **Q2)**\
> Thanks for this question as it gives us the opportunity to better distinguish our framing from relevant solutions. RL generalization, the deployment gap, and test-time adaptation share the goal of reducing performance degradation at deployment, but their assumptions about the underlying problem are fundamentally different from ours. All three assume that training and deployment are separate and distinct phases, each with their own static distributions. This assumption only stands if the underlying problem is non-continual. Our central argument is that measurable RL deployment is a continual RL problem and should be treated as one. [7] thoroughly covers the benefits of framing continual RL as the problem where the best agents never stop learning, meaning imposing a fixed train/test boundary is not a viable solution to the measurable deployment problem, but a structural mismatch with it. Applying solution methods like RL generalization, the deployment gap, or test-time adaptation to such setting forces a continual problem to be viewed as a series of non-continual ones, and can be viewed as working around the above mismatch, rather than resolving it.
>
> Our perspective is not an alternative framing of these existing problems, but one that challenges their core assumptions. Those approaches accept the train/test boundary as given and propose competent solutions within that setting. Our framing operates at a more fundamental level by questioning whether that static boundary should exist at all. We therefore argue that progress on deployment robustness requires not better solutions within the train-then-fix paradigm (as has historically been the field’s trend), but solutions designed while embracing the continual nature of the problem. We would strongly be in favour of highlighting this distinction further if accepted, and welcome reviewer suggestions beyond the aforementioned to do so.
>
> **Q3)**\
> We appreciate this suggestion. We agree more concrete grounding would strengthen the paper, and we see two ways to address this. First, we can expand our description of Cursor Tab's implementation with the additional technical details available to us. Second, we will point readers to existing small-scale experiment results that directly support our central claim (including [7]'s continual Q-learning experiments in switching MDPs, and [8]'s experiments showing agents that continually track outperform agents that converge in multiple small-scale environments) which together provide empirical grounding for the core argument. As a position paper, our primary goal is to reframe how the community thinks about deployment rather than to present novel empirical results. A more comprehensive empirical study remains as important future work, and we will note this explicitly if the paper is accepted.
>
> [1] Silver et al. 2021, Reward is Enough\
> [2] Everitt et al. 2017, Reinforcement Learning with a Corrupted Reward Channel\
> [3] Wang et al. 2020, Reinforcement learning with perturbed rewards\
> [4] Greenberg and Mannor 2021, Detecting Rewards Deterioration in Episodic Reinforcement Learning\
> [5] Leike et al. 2018, Scalable agent alignment via reward modeling: a research direction\
> [6] Abel et al. 2021, On the Expressivity of Markov Reward\
> [7] Abel et al. 2023, A Definition of Continual Reinforcement Learning\
> [8] Sutton et al. 2007, On the role of tracking in stationary environments\

---

> > ### Author Rebuttal · Reviewer_C5PX · 2026-04-02
> >
> > Thank the authors for their rebuttal. I raise my score.

---

### Official Review · Reviewer_mSf2 · 2026-03-20

**Significance:** 4
**Argument Clarity:** 3
**Rating:** 5
**Confidence:** 3

**Questions:**

1) While I am not particularly familiar with the recent literature on continual RL, e.g., Bowling et al., 2023, Abel et al., 2023, Elelimy et al., 2025, I am wondering what is the novelty of the position expressed in this paper with respect to what can be already inferred by the literature. In other words, can the authors comment on why the position paper is needed, beyond convincing the community to consider the recent works in continual RL?

2) I think the position could be more precise on the distinction between continual adaptation and repeated retrain of a deployed system. If one consider repeated retrain as a continual method, then one would argue that a high share of deployed RL agents are actually continual already.

3) One thing that is not explicitly discussed as alternative view is the following: Can we train an agent to be adaptable, so that it remains fixed at deployment? If we have a training set that spans all of the potential variations that can occur at deployment, we may be able to extract an history-based policy that adapts at deployment, even without changing the model parameters with additional data. This is essentially the premise of frameworks of Bayesian RL or meta RL.

If the authors can clarify these points, I will consider raising my score to accept.

**Alternative Views Section:**

Yes

**Compliance With Llm Reviewing Policy A Conservative:**

Affirmed.

**Discussion Potential:**

3

**Final Justification:**

See the rebuttal acknowledgement

**Paper Summary:**

The paper argues that deployed RL shall be designed to be continual, especially in the presence of a measurable deployment setting, in which a reward signal is available at deployment. The paper first recalls recent continual RL problem formulation, mostly from Abel et al. 2023, Bowling wt al. 2023, then states that measurable deployment is a continual RL problem, which coarsely means that a fixed policy cannot keep optimality in the long run. The paper then reports a few use cases where continual RL brought benefits to deployed agents. The call for actions is organized as a set of recommendations for researchers and practitioners.

**Position:**

Yes

**Position In Title:**

Yes

**Related Work:**

2

**Strengths And Weaknesses:**

Strengths
- The paper expresses a very relevant position at a time in which we are seeing usage of AI systems trained with RL soaring;
- The paper provides several insights and actionable recommendations to researchers and practitioners.

Weaknesses
- The position title sounds perhaps overly ambitious with respect to what is actually expressed in the paper, for which deployed RL is seen mostly as an opportunity than a necessity;
- The relation between continual RL and the train-then-fix paradigm tends to blur when re-training is factored in;
- The scope of the position is perhaps too wide, especially in the call for action: The paper provides a variety of recommendations that overhauls most of the processes to which research is currently conducted in RL. I am wondering whether splitting the position into more focused efforts, e.g., "deployable RL shall be evaluated with continual RL metrics" could result more effective in terms of community adoption. I am aware that this is highly personal and mostly based on taste.

Evaluation

I think the position expressed in this paper is relevant, significant, and timely for the RL community to hear. Thus, I am inclined for an accept recommendation. However, the scope of the position is very broad and the paper as-is may result overwhelming in terms of opinions for the reader to consider.

**Support:**

3

---

> ### Author Rebuttal · Authors · 2026-03-30
>
> We thank reviewer mSf2 for their insightful review. The strengths and weaknesses outlined are very relevant, and we appreciate their questions and interest in this line of work. We clarify the questions as follows:
>
>  **Q1)**\
> We appreciate this question, as it gets to the heart of our contribution. We agree that the broader argument for continual learning in the real world has been made before, and foundational works such as [1] and [2] have been instrumental in formalizing this insight.
>
> The novelty of our position lies not in reiterating that the real world deployment requires continual learning, but in identifying and characterizing the specific, practically prevalent deployment regime where a reward signal remains available to the finite, limited agent (i.e. measurable deployments, which are a specific subset of all possible “big” world settings). To our knowledge, this distinction did not previously exist in the CRL literature. This distinction allows practitioners to recognize when their deployment is a CRL problem and when it is not. Prior works did not provide the means to put this distinction into operation (which we provide through the introduction of the term “measurable deployment”, and our Call to Action).
>
> RL algorithms leverage evaluative feedback by design [3], yet the dominant practice is to discard this signal at deployment, reverting to fixed policies or human-driven retraining. As a CRL position paper, we aim to bridge the gap between a community that has theorized about continual learning and an industry that continues to treat measurable deployments as non-continual.
>
> **Q2)**\
> Using [4]’s definitions, repeated retraining is indeed a legitimate learning rule for continual adaptation (it is a valid map from a history to a distribution over the policy set). We agree it can constitute continual learning when it is integrated as part of the agent's internal operations, with the agent autonomously deciding when and how to update from its own experience, as mentioned on line 306 of section 4. Examples of this type of learning rule in the CRL literature include [5,6].
>
> In practice, however, retraining is almost always an external operation. It is triggered by human expertise rather than the agent's own history, with minimal to no agent control over timing, duration, or the data used. Under this framing, the agent can no longer be considered a continual learner. Therefore, most deployed RL agents are essentially fixed policies stitched together by human-initiated resets. We loosely point to this in Section 4, and we welcome suggestions for making it clearer. A secondary, practical distinction is efficiency: full retraining requires large datasets and MLOps pipelines retraining agents from scratch, whereas CRL solutions update selectively and incrementally. We will make sure to sharpen both points in the camera-ready version, if the paper is accepted.
>
>
> **Q3)**\
> This is an excellent deployment case to consider when discussing continual learning. Thanks for pointing it out. There are indeed environments where, by collecting enough data, one can represent all potential scenarios and state visits; therefore, with finite training, a model can learn everything there is to learn and perform well while fixed in deployment. We allude to these in line 409 of the alternative views, “Not all deployments need Continual Learning". For deployment settings where collecting such a training set is feasible, alternative methods could be used to deliver a performant policy that does not need to adapt further, as the reviewer mentions. Furthermore, as mentioned in Section 3, those settings wouldn’t be considered measurable deployments any longer, as the agent has the computational and/or representational capacity to perform competently with a finite training phase. We highlight, however, that, in practice, for many applications, collecting such a training set is not practical or possible, as there will always be cases where the training set is insufficient. We acknowledge that adding more references on meta-RL and Bayesian RL related to this view in the aforementioned alternative view on line 409 would strengthen the argument, and we will do so if the paper is accepted.
>
> Please do not hesitate to respond if any of these questions are not satisfactorily answered. We enjoyed engaging in this discussion and look forward to hearing the reviewer’s thoughts on this.
>
>
> [1] Abel et al. 2023, A definition of Continual Reinforcement Learning\
> [2] Javed and Sutton 2024, The Big World Hypothesis\
> [3] Sutton and Barto 2018, Reinforcement Learning: An Introduction\
> [4] Elelimy et al. 2025, Rethinking the foundations for continual reinforcement learning\
> [5] Dohare et al. 2024, Loss of plasticity in deep continual learning\
> [6] Hernandez-Garcia et al. 2025, Reinitializing weights vs units for maintaining plasticity in neural networks

---

> > ### Author Rebuttal · Reviewer_mSf2 · 2026-04-01
> >
> > I want to thank the authors for their thorough replies. I have now a better understanding on why this paper is needed, as well as more nuances on the relation between continual learning and re-training, and the measurable deployment setting.
> >
> > My concerns have been resolved and I am more convinced that this paper will spark a positive discussion at the conference. I am raising my score to accept.

---

### Decision · Program_Chairs · 2026-04-30

**Decision:**

Accept (regular)

**Comment:**

This paper argues that RL agents should be continually updated in settings where a reward signal is available (and the environment is potentially non-stationary). This position is argued and illustrated using case studies and a simple example.

One weakness of the paper is the Alternative Views section, where some of the alternatives seem too simple and are described using only a single sentence ("retraining is enough", "not all deployments need CL"). In the final version, the authors should discuss other approaches for handling non-stationary environments, including meta-learning and in-context learning.